# Identification and Evaluation of Sugarcane Cultivars for Antixenosis Resistance to the Leafhopper *Yamatotettix flavovittatus* Matsumura (Hemiptera: Cicadellidae)

**DOI:** 10.3390/plants13162299

**Published:** 2024-08-18

**Authors:** Jariya Roddee, Jureemart Wangkeeree, Yupa Hanboonsong

**Affiliations:** 1School of Crop Production Technology, Institute of Agricultural Technology, Suranaree University of Technology, 111 University Avenue, Nakhon Ratchasima 30000, Thailand; 2Innovation of Quality Enhancement of Agricultural Products for Agro-Industry-Research Center, Suranaree University of Technology, Nakhon Ratchasima 30000, Thailand; 3Department of Agricultural Technology, Faculty of Science and Technology, Thammasat University Rangsit Campus, Pathum Thani 10200, Thailand; 4Department of Entomology and Plant Pathology, Faculty of Agriculture, Khon Kaen University, Khon Kaen 40002, Thailand

**Keywords:** plant resistance, sugarcane cultivars, electrical penetration graph (EPG), honeydew, pest management

## Abstract

Understanding the settling preference, feeding behavior, honeydew production, and biophysical factors, such as trichome density, related to *Y. flavovittatus* leafhopper infestation in sugarcane cultivation is crucial for effective pest management strategies. This study investigated these aspects across nine sugarcane cultivars. Significant variability was observed among cultivars in terms of settling behavior, with KK3 and LK92-11 showing the highest number of settled leafhopper adults. Similarly, honeydew production varied significantly among cultivars, with KK3 and LK92-11 exhibiting the highest production. Employing the electrical penetration graph (EPG) technique provided insights into distinct probing behaviors across cultivars, highlighting correlations between settling preference, honeydew production, and specific EPG waveforms. Principal component analysis (PCA) categorized cultivars into four groups based on settling preference, honeydew production, feeding behavior, and biophysical factors. Strong correlations were found between settling preference, honeydew production, and various EPG waveforms, while negative correlations were observed with the number of silica cells and rows per unit area, indicating their potential role in deterring leafhopper settlement. We concluded that TPJ04-768 and K84-200 are promising for resistance against leafhoppers and, thereby, can be exploited in sugarcane breeding programs with regard to resistance against insects.

## 1. Introduction

*Yamatotettix flavovittatus* Matsumura (Hemiptera: Cicadellidae) is a phloem feeder that causes significant damage to sugarcane and transmits the sugarcane white leaf (SCWL) disease [1,2]. The extensive damage caused by leafhoppers in Thailand was initially observed in the Khon Kaen and Udonthanee provinces [3]. This leafhopper species consumes phloem sap on sugarcane through its stylets, making it a specialized herbivore. Both nymphs and adult leafhoppers primarily feed on sap at the base of tillers from phloem tissues, often undetected [1,2]. Additionally, the widespread use of insecticides to control leafhoppers not only increases crop production costs but also has negative environmental impacts, including potential pest outbreaks due to the destruction of natural enemies and the selection for pest resistance [4,5]. 

Population dynamic studies of *Y. flavovittatus* have indicated significant population surges during rainy seasons, particularly from July to August in Thailand, reaching their peak in August [6]. This period coincides with the highest prevalence of SCWL disease in sugarcane fields, typically observed from August to September. While conventional methods for controlling vector-borne diseases rely heavily on insecticide application in cultivated fields, they pose environmental risks and become ineffective in large-scale commercial sugarcane cultivation. Additionally, the emergence of insecticide-resistant strains further challenges the efficacy of chemical control. An alternative technique reported on controlling *Y. flavovittatus* involves the use of sticky traps to disrupt the insect vector’s dispersal behavior [7]. Furthermore, bacterial symbionts, particularly Wolbachia, have garnered attention for their potential role in insect population control. Studies on the population dynamic and phylogeny of Wolbachia in *Y. flavovittatus* have been conducted [8,9,10] and revealed its influence on the leafhopper’s life cycle and reproductive system [11].

Despite immense efforts, traditional management practices, such as crop covering with insect-proof screening, prove inadequate for effectively controlling insect vectors. Besides disease management through infected plant eradication, efficient methods for managing insect vectors and associated diseases have not been established. Host plant resistance emerges as a promising strategy for mitigating SCWL disease; however, identifying SCWL-resistant sugarcane varieties remains elusive. Advancements in developing resistant sugarcane varieties and exploring other potential management techniques are contingent upon a deeper understanding of *Y. flavovittatus* feeding behaviors [3].

Resistant sugarcane varieties are complementary in reducing insecticide use and promoting biological control for managing leafhoppers in tropical sugarcane. Host plant resistance is considered the most economical and efficient method for leafhopper management [12]. While *Saccharum officinarum* Linn is a host plant for leafhoppers, no detailed study has investigated the varieties in resistance against leafhoppers. Resistance in wild cultivars may be attributed to various morphological traits, such as plant surface characteristics (trichomes and silica cells), which can act as mechanical barriers for shelter, oviposition, and feeding [13,14,15,16,17]. No detailed study has been conducted to understand the bases of resistance in sugarcane cultivars and the role of various biophysical factors (trichome density, epicuticular wax content, and leaf sheath thickness) in imparting resistance against leafhoppers. 

The leafhopper *Y. flavovittatus* feeds on plant tissue by inserting its stylet bundle, along with a salivary sheath, into the plant, targeting the phloem tissue and regulating the ingestion of pressurized plant sap [3]. To investigate the feeding behaviors of this leafhopper, which serves as a vector for SCWL, researchers have employed the electrical penetration graph (EPG) technique [3,18,19]. Through EPG, the positioning of *Y. flavovittatus* stylets during probing can be determined, facilitating the correlation of feeding patterns with interactions within the plant tissue. The feeding process of the insect is characterized by specific waveforms [3], including non-probing (NP), stylet probing into epidermal cells (Yf1), stylet probing through mesophyll/parenchyma (Yf2), stylet contact with phloem and likely watery salivation (Yf3), active ingestion of sap from the phloem, probably sieve elements (Yf4) and unknown stylet activity in multiple cell types (Yf5). *Y. flavovittatus* exhibits a preference for ingesting phloem sap over other cell types, indicating its primary role as a phloem feeder [3]. The hypothesis posits that the duration of the sieve element phase is longer in susceptible sugarcane cultivars and shorter in resistant ones. This research aims to achieve two main objectives: firstly, to identify antixenosis resistance in sugarcane cultivars and investigate the potential influence of biophysical factors in resisting leafhopper feeding behavior and secondly, to evaluate the relative attractiveness of sugarcane cultivars across a spectrum of resistance levels, ranging from highly resistant to susceptible ones.

## 2. Results

### 2.1. Settling Preference of Y. flavovittatus Leafhopper Adults 

The settling behavior of the *Y. flavovittatus* leafhopper adults at 24 h after release varied significantly among cultivars (F_8,45_  =  73.90, *p*  ≤  0.001). Notably, the minimum number of *Y. flavovittatus* leafhopper adults settled significantly varied among cultivars, with K99-72 and TPJ04-768 showing distinct differences after release (Figure 1A). Among the cultivars, KK3 and LK92-11 exhibited the highest number of settled leafhopper adults, followed by KK07-250, K84-200, KK07-037, and K76-4. This trend persisted at 48 h (F_8,45_  =  68.23, *p*  ≤  0.001), 72 h (F_8,45_  =  56.32, *p* ≤  0.001), and 96 h (F_8,45_  =  64.76, *p*  ≤  0.001) post-release (Figure 1B,C), with maximum leafhopper adults settled observed on KK3 and LK92-11, while minimum leafhopper adults settled occurred on K84-200 and TPJ04-768 cultivars for 48 h, K99-72 and TPJ04-768 cultivars for 72 h, and TPJ04-768 cultivar for 96 h. 

### 2.2. Honeydew Production

The honeydew production area of the *Y. flavovittatus* leafhopper fed on different sugarcane cultivars (Appendix A) exhibited significant differences (*p* < 0.001), with the highest production observed on the KK3 and LK92-11 at 5 h and 10 h post-feeding. However, there were no significant differences in honeydew production area between leafhoppers that fed on KK07-250 and KK07-037 cultivars at 5 h and 10 h post-feeding. Moreover, TPJ04-768 showed the lowest honeydew production area at 5 h and 10 h (Figure 2).

### 2.3. Measuring Adult Leafhopper Feeding Behavior by EPG Technique 

#### 2.3.1. Comparison of the Frequency and Duration of EPG Waveforms

The EPG technique was employed to compare the frequency and duration of six probing waveforms (Np, Yf1, Yf2, Yf3, Yf4, and Yf5) [3] exhibited by the leafhopper *Y. flavovittatus* across nine sugarcane cultivars (Figure 3, Figure 4 and Figure 5). The six probing waveforms were observed in the leafhopper *Y. flavovittatus* for 10 h. Total probing duration (TPD), total waveform duration (TWD), waveform duration per event per insect (WDEI), and number of waveforms per event insect (NWEI) were calculated and compared between the nine sugarcane cultivars. 

#### 2.3.2. Total Probing Duration (TPD)

Quantitative analysis revealed significant variations in the total probing duration among the cultivars. The *Y. flavovittatus* exhibited the longest TPD on the KK3 cultivar (74.4%), followed by the K76-4 (54.6%) and K99-72 (46.0%). Meanwhile, the TPJ04-768 cultivar had the shortest TPD (Figure 3). 

**Figure 3 plants-13-02299-f003:**
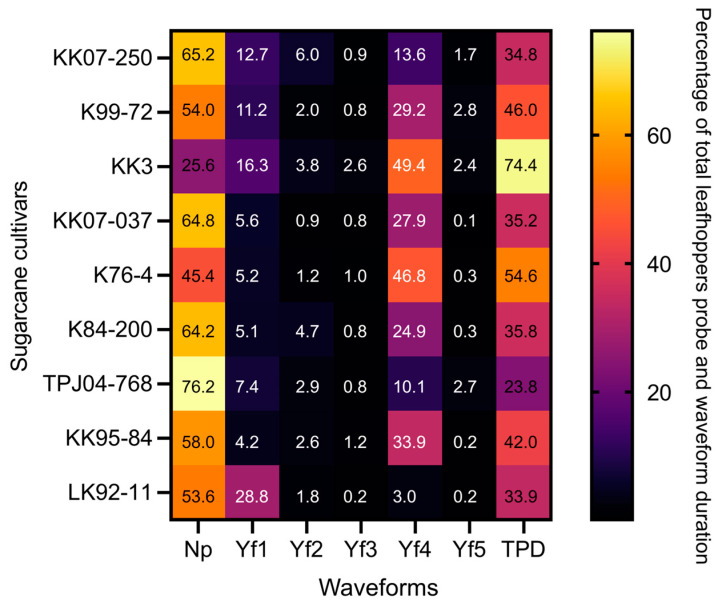
TPD (total probe duration) and TWD (total waveform duration) were recorded during the stylet probing behavior of *Yamatotettix flavovittatus* on nine sugarcane cultivars; each was a 10-h recording period. Waveform NP is non-probing, waveform Yf1 represents stylet probing into epidermal cells, waveform Yf2 represents stylet probing through mesophyll/parenchyma, waveform Yf3 represents stylet contact with phloem and likely watery salivation, waveform Yf4 stylet active ingestion of sap from the phloem, sieve elements, and waveform Yf5 represents stylet activity in multiple cell types.

#### 2.3.3. Total Waveform Duration (TWD)

Figure 3 depicts the distribution of total waveform duration across eight sugarcane cultivars. *Y. flavovittatus* leafhopper showed a high duration of non-probe (Np) and Yf4 waveform on eight sugarcane cultivars. Notably, on the TPJ04-768 cultivar, the *Y. flavovittatus* spent most of its time in Np waveform (76.2% of access time), followed by the KK07-037 (64.8%) and K84-200 (64.2%). While on the KK3 cultivar, the predominant waveform was Yf4 (49.4% of access time), which exceeded the K76-4 cultivar (46.8%) and the KK95-84 cultivar (33.9%) (Figure 3). The difference in TWD was statistically significant among cultivars for various waveforms, including Yf2 and Yf3, indicating variability in probing behavior (Figure 3).

#### 2.3.4. Waveform Duration per Event per Insect (WDEI) 

Analysis of waveform duration per event per insect (WDEI) revealed significant differences in the percentage of time spent on different waveforms across cultivars (Figure 4). All EPG activities varied significantly between the sugarcane cultivars except for pathway (waveform Yf1) (F_8,404_ = 2.87, *p* = 0.42) and stylet activity in multiple cell types (waveform Yf5) (F_8,30_ = 3.67, *p* = 0.20). The leafhopper *Y. flavovittatus* exhibited a high duration percentage of Np on LK92-11 and KK95-84 cultivars, which were 61.24% and 51.01%, respectively, and were significantly different from KK07-250, KK3, KK07-037, K76-4, K84-200, and TPJ04-768 cultivars (F_8,439_ = 4.36, *p* = 0.002). Yf2 predominated on KK07-250 and TPJ04-768 cultivars and differed significantly from other cultivars (F_8,139_ = 5.71, *p* = 0.045). In contrast, waveform Yf3 of leafhopper *Y. flavovittatus* showed a significantly higher duration percentage on KK07-037 than on other cultivars (F_8.97_ = 3.81, *p* = 0.016). Notably, the ingestion of sap from the phloem (waveform Yf4) lasted significantly longer over this period when leafhopper *Y. flavovittatus* fed on K76-4 (47.25%), KK07-037 (42.32%) K84-200 (38.18%) and KK3 cultivars (36.24%) than when they fed on other cultivars (F_8,236_ = 4.57, *p* = 0.034). The whole Section 2.3.4 refers to Figure 4.

**Figure 4 plants-13-02299-f004:**
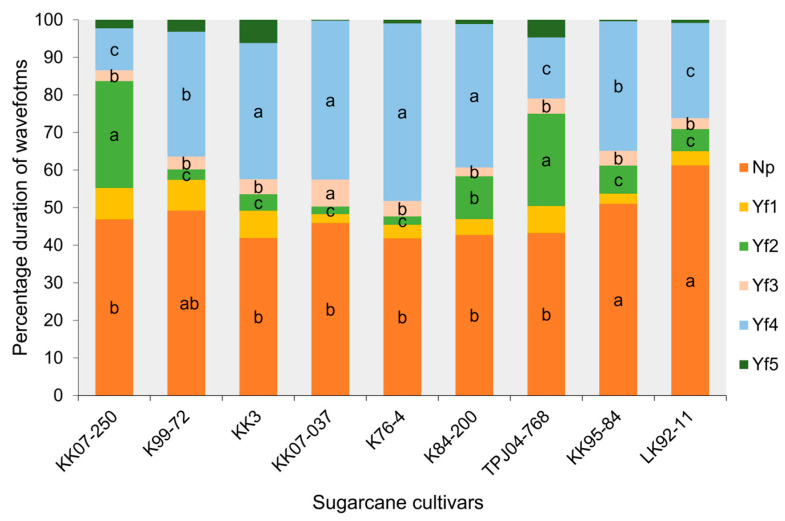
Comparison of different EPG waveform feeding patterns of leafhopper *Yamatotettix flavovittatus* on nine sugarcane cultivars for 10 h (average percentage of waveform duration per event per insect). The same letter is not significantly different (*p* > 0.05, Mann–Whitney U test and Tukey–Kramer test).

#### 2.3.5. Number of Waveforms per Event Insect (NWEI)

The frequency of waveform per event per insect (NWEI) varied significantly among cultivars for all EPG activities except for phloem salivation (waveform Yf3) (F_8,97_ = 6.89, *p* = 0.16) and stylet activity in multiple cell types (waveform Yf5) (F_8,30_ = 5.87, *p* = 0.70). Noteworthy differences were observed, such as a high frequency of Np on the TPJ04-768 cultivar, which was 50.49%, that significantly differed from other cultivars (F_8,439_ = 3.98, *p* = 0.003) (Figure 5). On the dominance of waveform Yf1, leafhopper *Y. flavovittatus* showed a high-frequency percentage on the KK07-037 cultivar, a difference that is significant from other cultivars (F_8,404_ = 4.64, *p* = 0.020). For waveform Yf2, leafhopper *Y. flavovittatus* showed a high-frequency percentage on the KK07-250 cultivar, which differed significantly from other cultivars (F_8,139_ = 2.57, *p* = 0.031). Additionally, the ingestion of sap from the phloem (waveform Yf4) occurred significantly and most frequently when the leafhopper *Y. flavovittatus* fed on LK92-11 (26.02%) and KK3 (25.77%), and it differed significantly from other cultivars (F_8,236_ = 3.21, *p* = 0.009). The whole Section 2.3.5 refers to Figure 5.

**Figure 5 plants-13-02299-f005:**
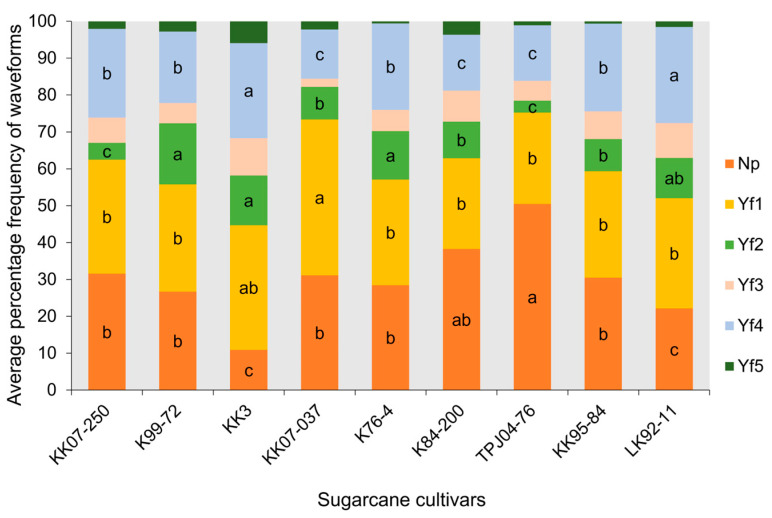
Comparison percentage of time for different EPG waveform feeding patterns of leafhopper *Yamatotettix flavovittatus* on nine sugarcane cultivars for 10 h (average percentage of waveform frequency per insect). The same letter is not significantly different (*p* > 0.05, Mann–Whitney U test and Tukey–Kramer test).

### 2.4. Variation in Biophysical Factors among Sugarcane Cultivars

This study examined the variability in biophysical factors (Figure 6 and Figure 7), including the number of silica cells per 100 μm^2^, the number of silica rows per 100 μm^2^, and trichome density among different sugarcane cultivars (Table 1). 

#### 2.4.1. The Number of Silica Cells per 100 μm^2^

Significant differences were observed in the number of silica cells per 100 μm^2^ among the cultivars (F_9,45_  =  40.35, *p* ≤ 0.001). The cultivars TPJ04-768 exhibited the highest number of silica cells, with a maximum of 143.00 cells per 100 μm^2^, followed by K99-72, which also showed a significantly higher count compared to other cultivars. Conversely, the cultivars KK3 displayed the lowest number of silica cells per 100 μm, which was only 18.49 cells/100 μm^2^. 

#### 2.4.2. Silica Row per 100 μm^2^

Similar to silica cells, significant variations were observed in the number of silica rows per 100 μm^2^ across cultivars (F_9,45_  =  24.87, *p* ≤ 0.001). The cultivars TPJ04-768 exhibited the highest number of silica rows, with a maximum of 10.40 rows per 100 μm^2^. Conversely, cultivars KK3, LK92-11, and KK95-84 exhibited the lowest number of silica rows per µm, with values of 5.4, 6.0, and 7.2 rows per 100 μm^2^, respectively. 

#### 2.4.3. Trichome Density 

Trichomes are hair-like structures on plant surfaces that can act as physical barriers against pests. Significant differences were observed in trichome density among the cultivars (F_9,45_  =  18.43, *p* ≤  0.001). Cultivars LK92-11 and KK3 had the highest trichome density, with values of 11.80 and 9.60, respectively. 

### 2.5. The Correlation between Settling Preference, Honeydew Production, Y. flavovittatus Feeding, and Biophysical Factors

A positive relationship was observed between the duration of settling preference and honeydew production. Specifically, the duration of settling preference increased, as did the honeydew production. Significant linear correlations were found between *Y. flavovittatus* feeding behavior and settling preference at different time intervals (24, 48, 72, and 96 h), as well as honeydew production at 5 h and 10 h. The correlation coefficients (r) ranged from 0.93 to 0.94, indicating strong positive associations between these variations (Figure 8). A settling preference at 24 h exhibited a positive correlation with the number of non-probe waveform events per insect. Furthermore, the settling preference at 72 h was a positive correlation with waveform Yf1 frequency events per insect (NWEI Yf1) and waveform Yf4 frequency events per insect (NWEI Yf4) (*r* = 0.86* and *r* = 0.87*, respectively). In addition, waveform YF4 frequency events per insect (NWEI Yf4) had a positive correlation with the settling preference at 24h and 72 h, waveform Yf1, Yf2 and Yf3 frequency events per insect (NWEI Yf1, Yf2, and Yf3) (*r* = 0.86*, *r* = 0.87*, *r* = 0.83*, *r* = 0.87* and *r* = 0.91*, respectively). 

In contrast, the number of silica cells exhibited strong negative correlations with settling preference at different time intervals (24 h, 48 h, 72 h, and 96 h), honeydew production (5 h and 10 h), and the number of waveforms Yf3 and Yf4 events per insect (*r* = −0.97*, *r* = −0.89*, *r* = −0.95*, *r* = −0.90*, *r* = −0.81*, *r* = −0.83*, *r* = −0.84* and, *r* = −0.90*, respectively) (Figure 8). Similarly, a negative correlation was observed between the number of silica rows and settling preference at 24 h and 72 h, honeydew production at 5 h and 10 h and the number of waveforms Yf3 and Yf4 events per insect (*r* = −0.86*, *r* = −0.82*, *r* = −0.85*, *r* = −0.86*, *r* = −0.81* and *r* = −0.88*, respectively) (Figure 8). These data indicate that cultivars with a high number of silica cells and a high density of silica rows may experience reduced leafhopper settling preference. 

### 2.6. Principal Component Analysis

The first two principal components (PCs) collectively accounted for 80.65% of the total data variability, with PC1 contributing 62.68% and PC2 contributing 17.97%. Through PCA, the sugarcane cultivars were categorized into four distinct groups (A, B, C, and D, Figure 9). Group A comprised the KK3 and LK92-11 cultivars, suggesting similarities in their characteristics related to settling preference, honeydew production, *Y. flavovittatus* feeding behavior, and biophysical factors. Group B included KK07-250, K99-72, and KK07-037 cultivars. Group C consisted of K76-4 and KK95-84 cultivars, and Group D included K84-200 and TPJ04-768 cultivars. 

## 3. Discussion

Host plant resistance is eco-friendly, safe, and generally compatible with other components of integrated pest management programs. Since leafhoppers are difficult to manage because of their feeding and migratory behavior, it is important to develop leafhopper-resistant varieties. In this study, we have characterized the feeding behavior of *Y. flavovittatus* using DC-EPG, honeydew production, settling preference, and biophysical factors data to screen nine sugarcane varieties, facilitating an efficient and detailed classification of sugarcane varieties for insect resistance.

### 3.1. The Settling Preference of Leafhopper 

The adults *Y. flavovittatus* among different cultivars revealed significant variations, as indicated by the analysis of variance (ANOVA). This highlights the importance of cultivar selection in influencing the behavior of leafhopper populations. Notably, the cultivar TPJ04-768 exhibited a minimum number of leafhopper adults settling after release, suggesting potential resistance or unattractiveness to these pests.

Among the cultivars studied, KK3 and LK92-11 consistently attracted the highest numbers of leafhoppers, both at 24 h and subsequently at 48 h, 72 h, and 96 h after release. This indicates a strong preference for these cultivars by *Y. flavovittatus*. The reasons for this preference could be attributed to various factors, such as plant morphology [17,20,21,22], chemical composition [23,24], or inherent genetic traits [25,26,27], that may make these cultivars more suitable or attractive to leafhoppers for settling and feeding. In contrast, cultivars such as K99-72, KK95-84, and KK99-72 consistently showed the lowest numbers of leafhoppers settling across the observation periods. Understanding the characteristics of these less-preferred cultivars could provide valuable insights into breeding programs to develop resistant or less-attractive plant varieties, thereby aiding in integrated pest management strategies. The observed differences in settling preference among cultivars underscore the complexity of plant-insect interactions and the need for tailored approaches in pest management. Farmers can make informed decisions regarding crop selection and cultivation practices to mitigate pest damage and optimize yields by identifying cultivars that are less susceptible to leafhopper infestation. Furthermore, this study emphasizes the significance of ongoing research efforts in breeding pest-resistant crop varieties as a sustainable solution to pest management challenges in agriculture.

### 3.2. The Honeydew Production of Leafhoppers

The leafhopper *Y. flavovittatus* feeding on various sugarcane cultivars revealed significant differences, highlighting the influence of cultivar selection on insect feeding behavior and physiology. The observed variations in honeydew production areas among different cultivars underscore the importance of understanding plant-insect interactions for effective pest management strategies in agriculture [28,29]. The highest honeydew production areas were recorded on cultivars KK3 and LK92-11 at 24h and 48 h after feeding initiation. This suggests that these cultivars may provide more favorable feeding conditions or nutrient content for *Y. flavovittatus*, increasing honeydew secretion. The factors contributing to this higher honeydew production could include plant physiology, nutrient availability, and possibly the presence of compounds that stimulate feeding in the sap-sucking insects [30,31,32,33]. Interestingly, cultivars KK07-250 and KK07-037 did not exhibit significant differences in honeydew production area compared to each other at 5 h and 10 h. This suggests these cultivars may share similar characteristics influencing leafhopper feeding behavior and subsequent honeydew production. In contrast, the cultivar TPJ04-768 consistently showed the lowest honeydew production area at 5 h and 10 h after feeding initiation. This indicates a potential deterrent effect or reduced suitability of this cultivar for *Y. flavovittatus* feeding and honeydew production. This may be attributed to changes in lipase activity, oil content, protein levels, and soluble sugars observed in previous studies, which affect the physiological and biochemical responses of cotton varieties resistant and susceptible to leaf curl virus (CLCuV) during germination and early seedling stages [30]. Further investigation into the biochemical and physiological factors underlying this reduced honeydew production on TPJ04-768 could elucidate mechanisms of resistance or non-preference against leafhopper infestation [30,34].

### 3.3. Stylet Penetration Behavior of Y. flavovittatus on the Sugarcane Cultivars

The electrical penetration graph technique provides valuable insights into the feeding behavior of insect pests and helps to determine the effects of plant antifeedants on the behavior of sap-sucking insects [35,36,37,38], such as the leafhopper *Y. flavovittatus* on different sugarcane cultivars. By comparing the frequency and duration of EPG waveforms, we can gain a better understanding of the interactions between the leafhoppers and the plant hosts, which is crucial for developing effective pest management strategies [3]. EPG analysis has explained some differences in the stylet penetration behavior of *Y. flavovittatus* on susceptible and resistant sugarcane cultivars. For instance, the total duration of EPG waveform related to phloem sap ingestion was much shorter on resistant sugarcane cultivars than on susceptible ones. In addition, the leafhopper *Y. flavovittatus* probed for longer periods on their respective susceptible cultivars than on resistant cultivars, and the leafhopper *Y. flavovittatus* spent a longer period in non-probed waveform in the resistant cultivars than on susceptible cultivars [17,37,38]. We identified six typical waveforms of leafhopper *Y. flavovittatus* with the help of a previously described histological study related to EPG and leafhopper’s stylet penetration regarding waveform classification [3,17,18]. Therefore, we chose their description as the main guide for EPG characterization. The TPD analysis revealed significant differences in the probing behavior of *Y. flavovittatus* among the sugarcane cultivars studied. Notably, the leafhoppers exhibited a longer TPD when feeding on the KK3 cultivar than other cultivars, indicating a prolonged feeding period. This suggests that KK3 may provide more favorable conditions for leafhopper feeding or that leafhoppers find it more attractive for probing and feeding activities. The TWD analysis showed variations in the duration of specific EPG waveforms across different sugarcane cultivars. For instance, the Np waveform, representing non-probing behavior, was observed for an extended period on TPJ04-768, indicating reduced probing activity on this cultivar. Similarly, the feeding activity of the planthopper biotype-3 *N. lugens* on five rice cultivars was monitored with an electronic recording device. The *N. lugens* probed repeatedly, salivated for a long time, and ingested for a very short period on resistant ‘IR56′ when compared with the susceptible cultivar [39]. Conversely, the Yf4 waveform, associated with ingestion of sap from the phloem, exhibited longer durations on cultivars such as K76-4, KK07-037, K84-200, and KK3, suggesting these cultivars may be more susceptible to leafhopper feeding on phloem sap [37,40,41].

The WDEI analysis further elucidated the feeding preferences of *Y. flavovittatus* among the sugarcane cultivars. Significant differences were observed in the duration of specific EPG waveforms, indicating varying levels of attractiveness or suitability of cultivars for leafhopper feeding. For instance, cultivars LK92-11 and KK3 showed the highest frequency of Yf4 waveform, suggesting a preference for feeding on phloem sap in these cultivars. The NWEI analysis highlighted differences in the frequency of EPG waveforms among the sugarcane cultivars. Cultivars TPJ04-768 and K84-200 exhibited higher frequencies of non-probe waveform, indicating reduced probing activity compared to other cultivars. Additionally, cultivars KK07-037 and KK07-250 showed higher frequencies of specific waveforms (Yf1 and Yf2, respectively), suggesting potential attractiveness to *Y. flavovittatus* for probing and feeding activities. Overall, these findings underscore the importance of cultivar selection in influencing the feeding behavior of *Y. flavovittatus* and subsequent pest damage in sugarcane crops. By understanding the preferences and behaviors of insect pests at the small components or a granular level using techniques like EPG, researchers and growers can develop targeted pest management strategies to minimize damage and optimize yields in sugarcane cultivation.

### 3.4. The Biological Sugarcane Cultivars Factor

The morphology and anatomical features of sugarcane cultivars play a significant role in influencing pest resistance and susceptibility. Here, we discuss the variations observed in the number of silica cells, silica rows, and trichome density among different cultivars, as highlighted in Table 1. The observed variations in silica cells, silica rows, and trichome density among sugarcane cultivars underscore the importance of plant anatomy in determining resistance or susceptibility to pests. Cultivars with higher densities of silica cells and rows may offer increased protection against pest infestation, potentially reducing the need for chemical interventions in pest management strategies. Further research into the mechanisms underlying these anatomical differences can inform breeding programs to develop pest-resistant sugarcane cultivars, ultimately contributing to sustainable agriculture practices. These efforts often target cultivars that exhibit antixenosis resistance. Antixenosis occurs when plant morphological or chemical factors adversely affect arthropod behavior, leading to delayed acceptance or rejection of a plant as a host [17,25,26,41,42]. Several previously studied glandular and non-glandular trichomes affect herbivores settling on host plants [43,44,45]. Specifically, the density of glandular trichomes is negatively correlated with whitefly attractiveness and oviposition per leaflet [46]. However, our study also revealed a negative correlation between lower trichome density and leafhopper attractiveness.

## 4. Materials and Methods

The experiments were conducted in the Entomology laboratory, and the study of biophysical factors of sugarcane cultivars was carried out using a scanning electron microscope (SEM) at the Synchrotron Light Research Institute (SLRI), Nakhon Ratchasima, Thailand.

### 4.1. Plant Materials

Nine sugarcane cultivars were evaluated in these experiments: Khon Kean07-250 (KK07-250), Khon Kean07-037 (KK07-037), Khon Kean3 (KK3), Khon Kean95-84 (KK95-84), Kasetsat99-72 (K99-72), Kasetsat76-4 (K76-4), Kasetsat84-200 (K84-200), Lampang-Kanchanaburi92-11 (LK92-11) (Office of the cane and sugar board) and TPJ04-768 (Japan International Research Center for Agricultural Sciences: JIRCAS). Cultivar KK3 was chosen as a susceptible reference cultivar for sugarcane white leaf disease [47], others were chosen because they were commercial sugarcane cultivars (LK92-11, K99-72), and some of them (KK07-250, KK07-037, KK95-84, KK07-037, KK95-84, K76-4 and K84-200) were in the selection final phase of the sugarcane inbreeding program. The auxiliary buds of nine sugarcane cultivars were cut and planted individually into 11 × 15 cm pots. The pots were kept for approximately 90 days in a greenhouse favorable to sugarcane growth with natural light photoperiod under insect-free conditions (including the test conditions parameters like temperature, relative humidity, light intensity, and duration of light). When the plants were 12 weeks old (five to six-leaf stage), they were then transported to the laboratory to test for free-choice or no-choice conditions and biophysical factors of sugarcane cultivars. These tests were performed with insects, and biophysical factors were used to refer to sugarcane traits.

### 4.2. Leafhoppers

Leafhopper *Yamatotettix flavovittatus* specimens were collected from sugarcane fields in Nakhon Ratchasima Province, northeast of Thailand (16°52′48.4″ N, 102°47′33.4″ E) using light traps (Blacklight) from 6:00 to 8:00 pm and maintained on the KK3 varieties of sugarcane in plastic pots. Leafhopper colonies were reared on sugarcane with a density of five females and five males per sugarcane pot in the laboratory conditions at 27 ± 2 °C with (12 h) light photoperiod. The five to six-leaf stage KK3 sugarcane plants were used as food for rearing the leafhopper and were replaced once every two weeks with fresh potted plants depending upon the insect population in rearing cages. The sugarcane pots were placed in rearing cages (68  ×  50  ×  50 cm) covered with nylon mesh, stitched along the four sides of the frame, and the top end tied with rope to avoid the escape of the leafhopper population. The 5 to 7-day-old adult leafhoppers were used for experiments. 

### 4.3. Settling Preference of Leafhopper Adults (Choice Test)

The preference of leafhopper adults towards different sugarcane cultivars was studied in a choice test condition; nine sugarcane cultivars were grown in plastic pots (11 cm diameter, 15 cm height). All plastic pots were placed randomly in a circle inside a net house or BugDorm-6E620 Insect Rearing Cage (W60 × D60 × H120 cm), MegaView Science Co., Ltd., Taichung, Taiwan. This test was repeated five times (a single potted plant represented one replicate) in a completely randomized design (CRD). Fifty pairs of adult leafhoppers were released in the center of the net house and allowed to freely select the plants (choice test) at the laboratory condition of 27 ± 2 °C. The number of male and female adults settled on the plants (stalks and leaves) was counted at 24, 48, 72, and 96 h after release.

### 4.4. Honeydew Production

The study employed a Completely Randomized Design (CRD) with 20 replications to compare honeydew production from the leafhoppers that fed on KK07-250, KK07-037, KK3, KK95-84, K99-72, K76-4, K84-200, LK92-11, and TPJ04-768 cultivars. The one-pair adult leafhopper (male and female) was released on the leaf cage per one-leaf sugarcane plant (second or third leaf). A 2-cm diameter filter paper (Whatman No. 1) (Appendix A) was placed on the bottom of the leaf cage. The leafhoppers were allowed to feed on nine sugarcane cultivars for 5 h and 10 h. After that, the insects were removed from the sugarcane leaf and then frozen in the −40 °C freezer. Each filter paper was treated with ninhydrin 0.1% to indicate if the honeydew was blue-rimmed. The blue spots represent phloem-derived honeydew. The filter paper was brought into a hot-air oven at 90 °C for 10 min, after which the filter papers were collected and photographed with a Nikon digital camera (Nikon Corp., Tokyo, Japan) on an illuminated bench. The area of all spots was measured from the images (Appendix A) using Image-J version 1.48 (National Institute of Health, Bethesda, MD, USA).

### 4.5. Electrical Penetration Graph (EPG) Technique 

The probing or feeding marks were used to assess the probing/feeding behavior of the leafhoppers *Y. flavovittatus* following the method of [3]. The nine-cultivar sugarcane test was grown in small pots (11 cm diameter, 15 cm height) with a single plant per pot in 20 replications (each pot represented one replicate). Leafhopper feeding behavior was recorded and classified using a GIGA-8 DC electrical penetration graph amplifier system introduced by [48,49]. Newly emerged female adults were starved for 1 h, cooled to −20 °C for 3 min, and then carefully connected to 2 cm length for 18.5 µm diameter gold wire (EPG system, Wageningen University) with conductive silver glue on insect prothorax. To complete the electronic circuit, leafhoppers were connected to the second or third leaf area of each sugarcane plant. The experiment was conducted in the Entomology laboratory room at 27 ± 2 °C with 60 ± 10% humidity under a 12 h light photoperiod. Probing behavior was continuously recorded for 10 h. At least 10 replicates per sugarcane cultivar were obtained. All recorded signals were analyzed using the probe 3.4 software version [50].

### 4.6. Biophysical Factors

Trichome density, number of silica cells, and rows were studied in 90-day-old (two to three-leaf stage) sugarcane plants to determine their potential role in antixenosis. These variables were analyzed and imaged under a Scanning Electron Microscope (SEM, FEI, Quanta450, Eindhoven, The Netherlands) at the Synchrotron Light Research Institute (SLRI), Nakhon Ratchasima, Thailand. Each living leaf was cut into 5 × 5 mm pieces, then fixed with 6% paraformaldehyde overnight and washed three times in phosphate-buffered saline (PBS). Dehydration of the leaf sheaths was conducted using different grades of ethanol (50, 70, 80, 95, and 100%), and then leaf sheath samples were allowed to dry in a desiccator under a vacuum overnight [51]. Samples were mounted on aluminum stubs using double-sided carbon tape and were sputter-coated with a thin layer of gold using an automated sputter coater. Finally, using secondary electron detectors, the leaf sheath specimens were observed under SEM at a magnification of 100 μm at an accelerating voltage of 15 kV. The density of trichomes and the number of silica cells and rows were calculated by counting the number per unit area. Each cultivar was examined in five replications (a leaf sheath sample from one plant represented one replicate).

### 4.7. Statistical Analysis

All statistical analyses were conducted using SAS program version 9.4 software (SAS Institute, Inc., Cary, NC, USA). Data from the settling preference experiment, calculated as the mean member of leafhopper per nine treatments at different time intervals, were analyzed separately for each 24, 48, 72, and 96 h after release and were subjected to the Tukey–Kramer at 0.05 significance level [32,51]. The average honeydew areas (mean ± standard error) were statistically compared among one another between the cultivars using the Tukey–Kramer at 0.05 significance level. Before analysis, the data’s normality and homogeneity of variance were checked. The data that did not follow normal distribution were transformed by square root. The probing behavior of *Y. flavovittatus* was calculated by the number of probes produced by leafhoppers and the total duration of the probing (Pr) phase versus the non-probing (NP) phase [18]. After measuring the typical waveforms related to *Y. flavovittatus* probing, the following variables (mean ± standard error) were calculated, as described in [52]: the total probing duration (TPD), total waveform duration (TWD), waveform duration per event per insect (WDEI) and the number of waveform event per insect (NWEI). Variable values over a 10-h access period were statistically compared between the sugarcane variety for six waveforms: NP, Yf1, Yf2, Yf3, Yf4, and Yf5 using the Stylet program with its non-parametric analyses [3]. All summarising statistics were produced using Excel. EPG variables (mean ± standard error) were statistically compared among one another within the cultivar using the Tukey–Kramer at *p* < 0.05, as the SAS program Backus v. 2.0 described in [52]. The mean was compared using the Tukey–Kramer test for trichome density, number of silica cells, and rows response variables. 

For correlation analysis (Pearson correlation), [32,53] was conducted on mean values of settling preference, honeydew production, *Y. flavovittatus* feeding behavior, and biophysical factors to identify the relationships between parameters in this study. Differences were considered significant at a probability level of 5%. Principal component analysis (PCA) was used to evaluate the relationship between nine sugarcane cultivars. PCA was employed as a multivariate statistical method to discern patterns among the investigated sugarcane cultivars based on multiple parameters, including settling preference, honeydew production, *Y. flavovittatus* feeding behavior, and biophysical factors. PCA was carried out using Origin Pro 2024 software (OriginLab Corporation, Northampton, MA, USA) [54] to obtain the relationship between the sugarcane cultivar and 18 from 21 parameters (Appendix A).

## 5. Conclusions

The findings emphasize the significance of selecting cultivars with traits that deter or minimize pest feeding and honeydew production for sustainable pest management in sugarcane cultivation. By identifying cultivars that are less preferred by leafhoppers and result in lower honeydew production, farmers can make informed decisions regarding crop selection and integrated pest management practices to mitigate pest damage and optimize yields. Moreover, this study highlights the importance of ongoing research efforts aimed at understanding plant-insect interactions and developing pest-resistant crop varieties to address pest management challenges in agriculture. Based on the experiments carried out, TPJ04-768 and K84-200 are considered promising for resistance to leafhoppers and may be exploited in sugarcane breeding programs for resistance to insects.

## Figures and Tables

**Figure 1 plants-13-02299-f001:**
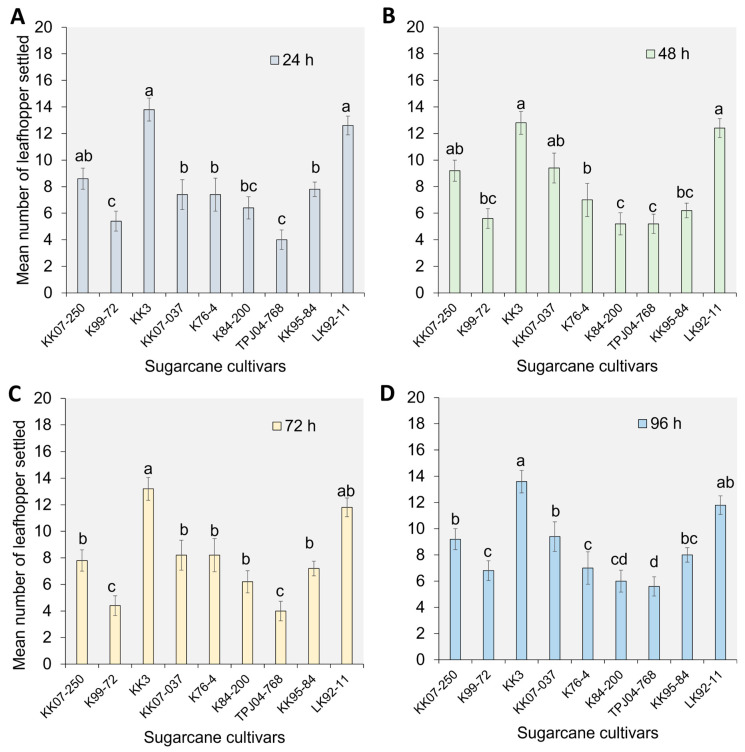
The number of *Yamatotettix flavovittatus* leafhopper adults settled on sugarcane plants. Different letters represent significant differences at *p* = 0.05 (Tukey’s test) level for each day after release separately; the bar represents the standard error. (**A**–**D**); The number of *Yamatotettix flavovittatus* leafhopper adults settled on sugarcane plants at 24, 48, 72, and 96, respectively.

**Figure 2 plants-13-02299-f002:**
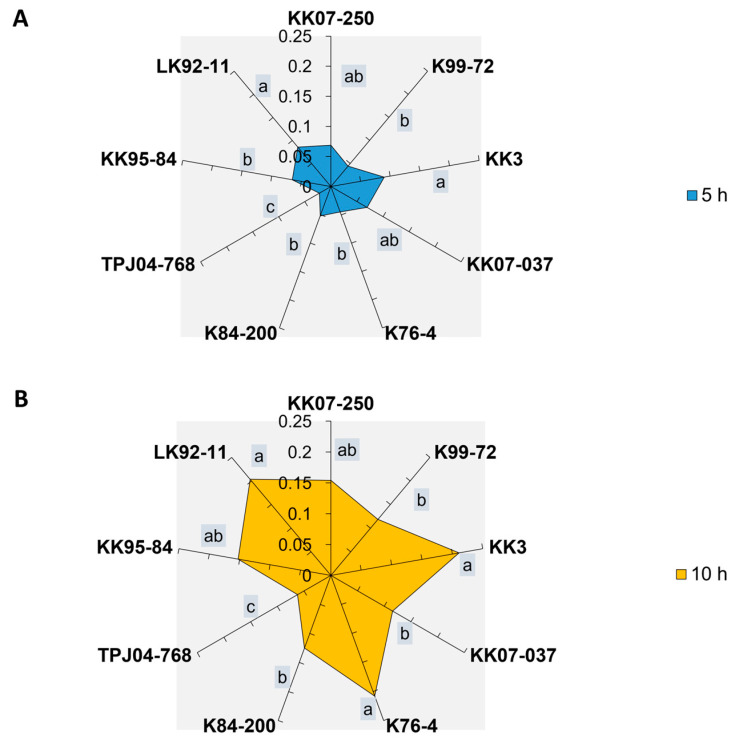
The honeydew area (mm^2^) of leafhopper *Yamatotettix flavovittatus* on nine sugarcane cultivars. Mean ± SEM with the same letter at the top category is not significantly different at *p* = 0.05 (Tukey’s test). (**A**) honeydew area of leafhopper at 5 h. (**B**) honeydew area of leafhopper at 10 h.

**Figure 6 plants-13-02299-f006:**
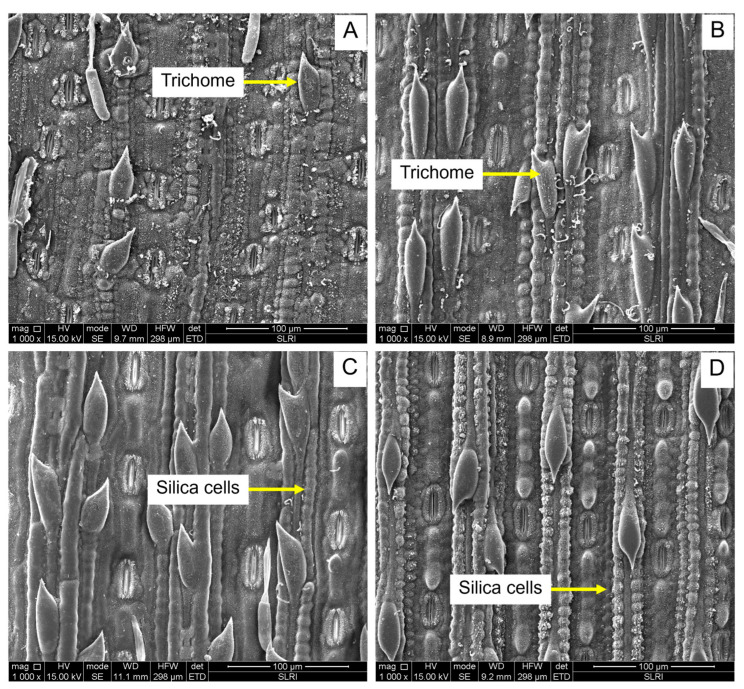
Scanning electron micrograph (SEM) of trichome density and silica cells of sugarcane cultivars. (**A**) Ultrastructure of trichomes of K84-200; (**B**) Ultrastructure of trichomes of LK92-11; (**C**) Ultrastructure of silica cells of susceptible, KK3; (**D**) Ultrastructure of silica cells of KK95-84.

**Figure 7 plants-13-02299-f007:**
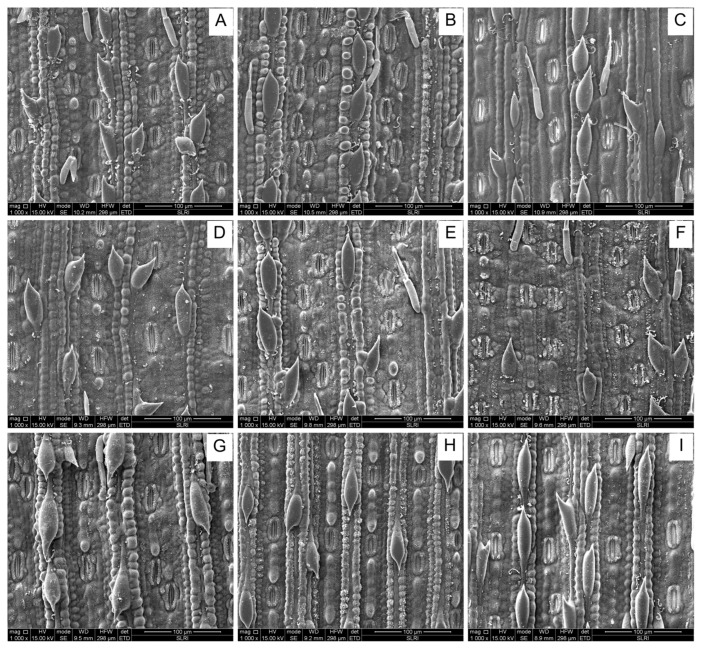
Scanning electron micrograph (SEM) of trichome density and silica cells of nine sugarcane cultivars. (**A**) KK07-250 (**B**) K99-72 (**C**) KK3 (**D**) KK07-037 (**E**) K76-4 (**F**) K84-200 (**G**) TPJ04-768 (**H**) KK95-84 (**I**) LK92-11.

**Figure 8 plants-13-02299-f008:**
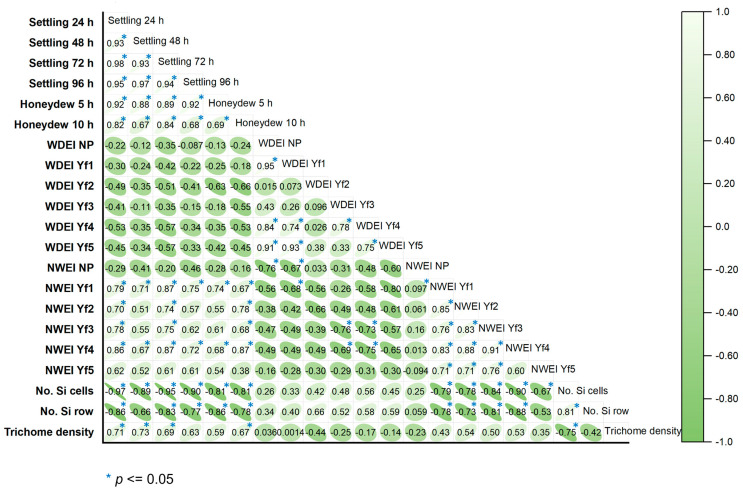
Correlation coefficients (Pearson correlation) and significance levels of settling preference, honeydew production, *Yamatotettix flavovittatus* feeding, and biophysical factors characters among nine sugarcane varieties. * Significant at 0.05 probability level.

**Figure 9 plants-13-02299-f009:**
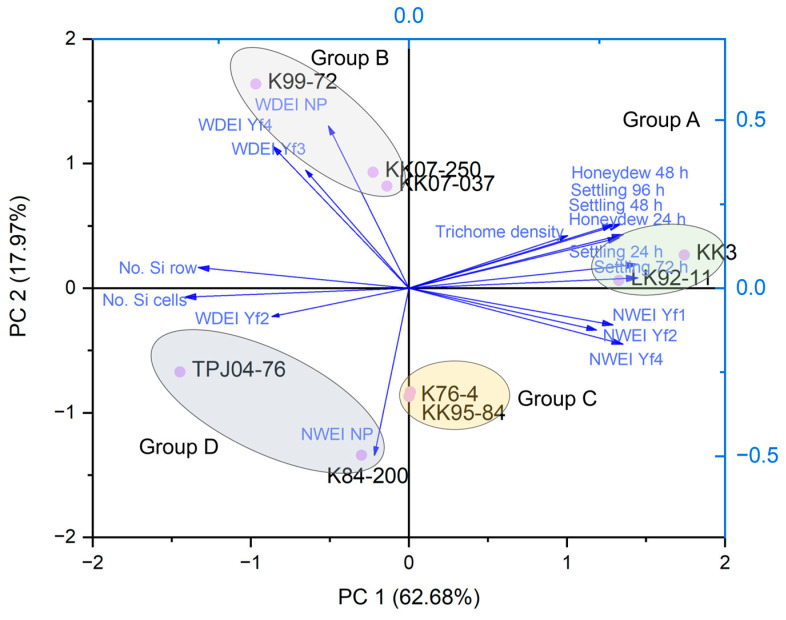
Principal component analysis (PC1 and PC2) of nine sugarcane cultivars based on the settling preference, honeydew production, *Yamatotettix flavovittatus* feeding behavior, and biophysical factors.

**Table 1 plants-13-02299-t001:** Biophysical factors of sugarcane cultivars.

Sugarcane Cultivars	No. Silica Cells per 100 µm^2^				No. Silica Rows per 100 µm^2^				Trichome Density per 100 µm^2^			
KK07-250	109.2	±	8.40	bc ^1/^	8.4	±	0.93	ab	8.6	±	0.93	ab
K99-72	132.4	±	6.52	ab	9.0	±	1.94	a	8.4	±	0.92	b
KK3	18.5	±	3.79	d	5.4	±	0.92	c	9.6	±	1.63	a
KK07-037	119.6	±	3.76	b	8.4	±	0.58	ab	7.8	±	0.58	c
K76-4	99.6	±	3.87	c	8.2	±	0.37	b	8.8	±	0.37	ab
K84-200	118.8	±	5.00	b	8.0	±	0.80	b	8.2	±	0.80	b
TPJ04-768	143.0	±	11.65	a	10.4	±	0.73	a	6.2	±	0.73	c
KK95-84	115.2	±	4.22	b	7.2	±	0.92	c	5.6	±	0.81	c
LK92-11	27.5	±	1.58	d	6.0	±	0.81	c	11.8	±	1.63	a

^1/^ mean ± SEM with the same letter in each collum category is not significantly different at *p* = 0.05 (Tukey’s).

## Data Availability

Data can be provided upon request from the lead author.

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
