# Peer review of "Identification and Evaluation of Sugarcane Cultivars for Antixenosis Resistance to the Leafhopper Yamatotettix flavovittatus Matsumura (Hemiptera: Cicadellidae)"

_plants, 2024, doi:10.3390/plants13162299_

Round 1

Reviewer 1 Report

Comments and Suggestions for Authors

Please carefully revise the manuscript before resubmission. See some of my comments below. 

In this research study, the authors delve into several critical aspects related to Y. flavovittatus leafhopper infestation in nine distinct sugarcane cultivars. The investigated factors include settling preference, feeding behavior, honeydew production, and relevant biophysical parameters. The findings from this study hold significant implications for future efforts in breeding resistant sugarcane varieties and effectively managing the sugarcane white leaf disease.

Major concerns

The manuscript has many careless mistakes and should be revised carefully before I can make suggestions for further improvements. The data provided in the figures are not correctly analyzed in the text. Having such mistakes in the manuscript suggests that the authors did not put a serious effort in writing the manuscript. And that reflects poorly on the credibility of the conclusions they make. Here are some examples:

1.       On lines 107-108, the authors mention, “minimum leaf hopper adults settled occurred on K99-72, KK95-84 and KK99-72 cultivars”. But in Figure 1, KK95-84 does not show minimum settling. Instead, TPJO4-768 shows minimum settling. There is also a typo since K99-72 and KK99-72 are the same cultivar.

2.       Line 144 mentions “Figure 3 depicts the distribution of total waveform duration across eight sugarcane cultivars. But the Figure 3 y-axis shows the percentage of leafhoppers having a certain EPG waveforms. Which one is correct?

Another error is in the explanation of the TPD abbreviation. In the caption of Figure 3, the authors explain TPD as (percentage of recording time). But the correct explanation in the other parts of the manuscript is (Total probing duration).

In addition, Figure 3 should include statistics like the other figures to make the results more valuable.

3.       On lines 178-179, the authors mention, “a high frequency of Np on TPJ04-768, and K84- 200 cultivars were significantly different from other cultivars”. But in Figure 5, the statistics show that K84- 200 is not significantly different than KK95-84, K76-4, KK07-037, K99-72, and KK07-250 as all have the letter “b”. There is also a typo in the figure TPJ04-76 should be TPJ04-768.

Comments on the Quality of English Language

The quality of the english language is fine. 

Author Response

Comments 1: On lines 107-108, the authors mention, “minimum leaf hopper adults settled occurred on K99-72, KK95-84 and KK99-72 cultivars”. But in Figure 1, KK95-84 does not show minimum settling. Instead, TPJO4-768 shows minimum settling. There is also a typo since K99-72 and KK99-72 are the same cultivar. 

Response 1: Thank you for pointing this out. We agree with this comment. Therefore, we revised to while minimum leafhopper adults settled occurred on K84-200 and TPJ04-768 cultivars for 48 h, K99-72, and TPJ04-768 cultivars for 72 h and TPJ04-768 cultivars for 96 h on page 3 lines 108-109.

Comments 2: Line 144 mentions “Figure 3 depicts the distribution of total waveform duration across eight sugarcane cultivars. But Figure 3 y-axis shows the percentage of leafhoppers having a certain EPG waveforms. Which one is correct?

Another error is in the explanation of the TPD abbreviation. In the caption of Figure 3, the authors explain TPD as (percentage of recording time). But the correct explanation in the other parts of the manuscript is (Total probing duration).

In addition, Figure 3 should include statistics like the other figures to make the results more valuable.

Response 2: Agree. We have modified Figure 3 and changed the TPD (percentage of recording time) to total probe duration on page 5, line 140.

Comments 3: On lines 178-179, the authors mention, “a high frequency of Np on TPJ04-768, and K84- 200 cultivars were significantly different from other cultivars.” But in Figure 5, the statistics show that K84- 200 is not significantly different from KK95-84, K76-4, KK07-037, K99-72, and KK07-250, as all have the letter “b.” There is also a typo in the figure: TPJ04-76 should be TPJ04-768.

Response 3: Thank you for pointing this out. We agree with this comment. Therefore, we revised 

to a high frequency of Np on the TPJ04-768 cultivar significantly differed from other cultivars (F8,439 = 3.98, p = 0.003) was 50.49% (Figure 5) on page 6 lines 182-184.

Reviewer 2 Report

Comments and Suggestions for Authors

Title: Identification and Evaluation of Sugarcane Cultivars for Antixenosis Resistance to the Leafhopper Yamatotettix flavovittatus Matsumura (Hemiptera: Cicadellidae)

Following corrections are suggested to improve the presentation

Minor corrections

Line 16, Delete sugarcane white leaf (SCWL) disease

Line 99 & 321, delete underline for scientific name of the leafhopper

Line 133-134, KKS cultivar (74.4%)

Line 396-397, and others were chosen because they were commercial sugarcane cultivars (list the varieties) and some of them (list the varieties) were

Line 401, natural light photoperiod under insect-free conditions (include the test conditions parameters like temperature, relative humidity, light intensity and duration of light). 

Author Response

Comments 1: Line 16, Delete sugarcane white leaf (SCWL) disease

Response 1: Thank you for pointing this out. We have deleted the sugarcane white leaf (SCWL) disease on page 1 line 16

Comments 2: Lines 99 & 321, delete underline for the scientific name of the leafhopper

Response 2: Agree. We have deleted the underline for the scientific name of the leafhopper on page 3 line 99 and page 12 line 328

Comments 3: Line 133-134, KKS cultivar (74.4%)

Response 3: Sorry, we did not find the mistake in the manuscript following the reviewer's suggestion. The Y. flavovittatus exhibited the longest TPD on KK3 cultivars (74.4%), followed by the K76-4 (54.6%) and K99-72(46.0%).

Comments 4: Line 396-397, and others were chosen because they were commercial sugarcane cultivars (list the varieties), and some of them (list the varieties) were

Response 4: Thank you for pointing this out. We have added a list of the varieties of sugarcane on page 14 lines 407-408 chosen because they were commercial sugarcane cultivars (LK92-11, K99-72), and some of them (KK07-250, KK07-037, KK95-84, KK07-037, KK95-84, K76-4, and K84-200)

Comments 5: Line 401, natural light photoperiod under insect-free conditions (include the test conditions parameters like temperature, relative humidity, light intensity, and duration of light). 

Response 5: Agree. We have added the test conditions parameters (like temperature, relative humidity, light intensity, and duration of light) on page 14, lines 412-413.

Reviewer 3 Report

Comments and Suggestions for Authors

The manuscript plants-3038990 entitled “Identification and Evaluation of Sugarcane Cultivars for Antixenosis Resistance to the Leafhopper Yamatotettix flavovittatus Matsumura (Hemiptera: Cicadellidae)” examined resistance/susceptibility of nine sugarcane cultivars to a pest insect. Based on settling preference, honeydew production and feeding behavior of leafhopper two resistant cultivars were identified. Resistance of sugarcane cultivars were also related to trichome density, number of silica cells and rows per unit of area. Resistant cultivars are less preferred, leafhoppers produce less honeydew and their feeding is minimized. TPJ04-768 cultivar had the highest number of silica cells and rows which were negatively correlated with settling and honeydew production. Such researches are important since pest-resistant cultivars represent sustainable mean of insect pest control. My comments and suggestions are listed below.

ABSTRACT

-Line 16: The first sentence is not complete.

-Line 16-18: Consider replacing “affecting” with “related to”. Also, explain here which biophysical factors you examined.

INTRODUCTION

-Line 75: Mention several biophysical factors in parentheses.

RESULTS

-Line 105: Put “h” after “48”.

-FIGURE 1: There is no abbreviation DAR in figures that should be explained. Therefore, you can delete DAR. After “Tukey’s” insert “test” or “post hoc test”.

-FIGURE 2: Replace “Bars” with “Values”. After “Tukey’s” insert “test” or “post hoc test”.

-Lines 129-130: I suggest you to explain abbreviations here.

-Line 134: Mention that TPJ04-768 had the shortest TPD.

-FIGURE 3: Replace “percentage of recording time” “with “total probing duration”. TWD is not in the figure. So, you should explain how it is presented in the figure.

-Line 152: Does it refer to Figure 4?

-Line 158: You can delete “no significant different between the nine sugarcane cultivars”.

-Lines 159-161: The sentence is confused and not grammatically correct. These cultivars are not significantly different from K99-72.

-Lines 161-164: Change into “Yf2, predominated on KK07-250 and TPJ04-768 cultivars and differred significantly from other cultivars……”; “In contrast, waveform Yf3 of leafhopper Y. flavovittatus shows significantly higher duration percentage on KK07-037 than on other cultivars …….”

-Lines 164-168: Change into “Notably, the ingestion of sap from the phloem (waveform Yf4) lasted significantly longer over this period when leafhopper Y. flavovittatus fed on K76-4 (47.25%), KK07-037 (42.32%), K84-200 (38.18%) and KK3 cultivars (36.24%) than on other cultivars (F8,236 = 4.57, p = 0.034).” In the whole 2.3.4. section refer to Figure 4.

-Lines 173, 192, 207: Add “test” after “Kramer”.

-Section 2.3.5.: Please, correct grammatical errors in lines 177-182 similarly to my corrections for section 2.3.4.

-Line 194: In Material and methods, you wrote “….number of silica cells and rows were calculated by counting the number per unit area.” Is “100 um” correct or it is “um2”? In Table 1 put “rows” instead of “row”.

-Line 233: Correlation 0.09 is not strong.

-FIGURE 9: Traits overlap. Can you replace trait names with numbers and explain the numbers in caption to figure? I believe that figure will be more clear.

DISCUSSION

-Line 286: Replace “this” with “the”.

-Lines 317-320: In this sentence, you described your results. So, you should explained the connection with cited references.

-The similar comment as for lines 317-320 can be given for lines 331-334 and 346-349.

-Line 363: EPG is not molecular technique.

-Line 380: Numbers of cited references should be in increasing order. Check this in the whole text.

MATERIAL AND METHODS

-Lines 401-403: Please, rewrite the sentence. What is “free-choice no-choice conditions”? Maybe you can put “and” or “/” between these two tests. Mention that these tests were performed with insects and that biophysical factors refer to sugarcane traits.

-Line 478: I suppose that average values were compared between (not within) cultivars.

REFERENCES

-Line 574: Replace “Matsumuratettixhiroglyphicus” with “Matsumuratettix hiroglyphicus”.

Comments on the Quality of English Language

I suggested corrections in my comments to authors. 

Author Response

Comments 1: Please revise the abstract, there are some mistakes there, quite at the beginning

Response 1: Thank you for pointing this out. We have deleted the first sentence on page 1 line 16

Comments 2: The leafhoppers were collected from fields for rearing- I assume it was just their offspring that served to conduct the study? Where larvae put on the studied plants or just adult specimens?

Response 2: We used the 5–to 7-day-old adult leafhoppers. We added the sentence “The 5–to 7-day-old adult leafhoppers were used for experiments.” On page 14 line 429.

Comments 3: It is generally important to highlight whether Y. flavovittatus is monophagous on sugarcane or if it may utilise also other plant species.

Response 3: The Y. flavovittatus is monophagous on sugarcane. Previous studies have shown that leafhoppers exclusively feed on a single plant species and can transmit phytoplasma only in sugarcane (Hanboonsong et al., 2002; 2006).

Comments 4: Why K84-200 was not clustered with group C? it is closer to it and not to TPJ04-76 – is there any statistical evidence for that?

Response 4: Although the K84-200 cultivar is closer to group C, it was clustered with group D because the principal components analysis showed a negative correlation with the parameters for K84-200, whereas group C had a positive correlation.

Comments 5: Line 174 – should be: per event per insect? 

Response 5: Thank you for pointing this out. We have changed it to “per event per insect”

Reviewer 4 Report

Comments and Suggestions for Authors

This an interesting manuscript on the very important problem of managing pests. Phloem-feeding bugs pose a great threat to cultivated plants. Therefore, any studies on this subject are crucial for proper management. I have some issues concerning the manuscript:

1)                  Please revise the abstract, there are some mistakes there, quite at the beginning

2)                  The leafhoppers were collected from fields for rearing- I assume it was just their offspring that served to conduct the study? Where larvae put on the studied plants or just adult specimens?

3)                  It is generally important to highlight whether Y. flavovittatus is monophagous on sugarcane or if it may utilise also other plant species.

4)                  Why K84-200 was not clustered with group C? it is closer to it and not to TPJ04-76 – is there any statistical evidence for that?

5)                  Line 174 – should be: per event per insect? 

Author Response

ABSTRACT

Comments 1: -Line 16: The first sentence is not complete.

Response 1: Thank you for pointing this out. We have deleted the first sentence on page 1 line 16

Comments 2: -Line 16-18: Consider replacing “affecting” with “related to”. Also, explain here which biophysical factors you examined.

Response 2: Agree. We have replaced “affecting” with “related to” on page 1 line 17.

INTRODUCTION

Comments 3: -Line 75: Mention several biophysical factors in parentheses.

Response 3: Agree. We added the various biophysical factors (trichome density, epicuticular wax content, and leaf sheath thickness) on page 2 lines 74-75.

RESULTS

Comments 4 -Line 105: Put “h” after “48”.

Response 4: Agree. We added “h” after “48” on page 3 line 105.

Comments 5-FIGURE 1: There is no abbreviation DAR in figures that should be explained. Therefore, you can delete DAR. After “Tukey’s” insert “test” or “post hoc test”.

Response 5: Agree. We delete DAR and insert “test” to “different letters representing significant differences at P = 0.05 (Tukey’s test) level for each day after release separately; the bar represents the standard error” in Figure 1, lines 112-113.

Comments 6-FIGURE 2: Replace “Bars” with “Values”. After “Tukey’s” insert “test” or “post hoc test”.

Response 6: Agree. We Replace “Bars” with “Values and “Tukey’s” insert “test” in Figure 2 lines 123-124.

Comments 7-Lines 129-130: I suggest you to explain abbreviations here.

Response 7: Agree. We have explained the abbreviation “Total probing duration (TPD), Total waveform duration (TWD), Waveform duration per event per insect (WDEI), and Number of waveforms per event insect (NWEI)” on page 4, lines 130-132.

Comments 8-Line 134: Mention that TPJ04-768 had the shortest TPD.

Response 8: Agree. We have mentioned that the TPJ04-768 cultivar had the shortest TPD on page 5 lines 137-138.

Comments 9-FIGURE 3: Replace “percentage of recording time” “with “total probing duration”. TWD is not in the figure. So, you should explain how it is presented in the figure.

Response 9: Agree. We have replaced “percentage of recording time” “with “total probing duration” in Figure 3. The TWD is the total waveform duration calculated as a percentage of waveform duration shown in the figure.

Comments 10-Line 152: Does it refer to Figure 4?

Response 10: TWD refers to Figure 3. We have mentioned Figure 3 on line 156.

Comments 11-Line 158: You can delete “no significant different between the nine sugarcane cultivars”.

Response 11: Agree. We deleted “no significant different between the nine sugarcane cultivars” on page 5 line 162

Comments 12-Lines 159-161: The sentence is confused and not grammatically correct. These cultivars are not significantly different from K99-72.

Response 12: Agree. We have rewritten the sentence to “The leafhopper Y. flavovittatus exhibited a high duration percentage of Np on LK92-11 and KK95-84 cultivars were significantly different from KK07-250, KK3, KK07-037, K76-4, K84-200, and TPJ04-768 cultivars” on page 5 line 164.

Comments 13-Lines 161-164: Change into “Yf2, predominated on KK07-250 and TPJ04-768 cultivars and differred significantly from other cultivars……”; “In contrast, waveform Yf3 of leafhopper Y. flavovittatus shows significantly higher duration percentage on KK07-037 than on other cultivars …….”

Response 13: Agree. We have rewritten the sentence to “Yf2 predominated on KK07-250 and TPJ04-768 cultivars and differed significantly from other cultivars (F8,139 = 5.71, p = 0.045). In contrast, waveform Yf3 of leafhopper Y. flavovittatus shows a significantly higher duration percentage on KK07-037 than on other cultivars (F8,97 = 3.81, p = 0.016).” on pages 5-6, lines 165-168.

Comments 14-Lines 164-168: Change into “Notably, the ingestion of sap from the phloem (waveform Yf4) lasted significantly longer over this period when leafhopper Y. flavovittatus fed on K76-4 (47.25%), KK07-037 (42.32%), K84-200 (38.18%) and KK3 cultivars (36.24%) than on other cultivars (F8,236 = 4.57, p = 0.034).” In the whole 2.3.4. section refer to Figure 4.

Response 14: Agree. We have rewritten the sentence. On page 6 lines 168-172.

Comments 15-Lines 173, 192, 207: Add “test” after “Kramer”.

Response 15: Agree. We added “test” after “Kramer”. Line 177

Comments 16-Section 2.3.5.: Please, correct grammatical errors in lines 177-182 similarly to my corrections for section 2.3.4.

Response 16: Agree. We have corrected grammatical errors in section 2.3.5. on page 6.

Comments 17-Line 194: In Material and methods, you wrote “….number of silica cells and rows were calculated by counting the number per unit area.” Is “100 um” correct or it is “um2”? In Table 1 put “rows” instead of “row”.

Response 17: Agree. We have changed to μm2 and 1 put “rows” instead of “row” in Table 1

Comments 18-Line 233: Correlation 0.09 is not strong.

Response 18: Agree. We have changed to 0.93 on page 9 line 237.

Comments 19-FIGURE 9: Traits overlap. Can you replace trait names with numbers and explain the numbers in caption to figure? I believe that figure will be more clear.

Response 19: We have tried replacing trait names with numbers, but it is no clearer. However, we have designed the Figure so that the traits do not overlap (Figure 9).

DISCUSSION

Comments 20-Line 286: Replace “this” with “the”.

Response 20: Agree. We replaced “this” with “the” on page 11 line 289.

Comments 21-Lines 317-320: In this sentence, you described your results. So, you should explained the connection with cited references.

Response 21: We added the discussion on page 12 lines 321-324.

Comments 22-The similar comment as for lines 317-320 can be given for lines 331-334 and 346-349.

Response 22: We added the discussion on page 13 lines 353-356.

Comments 23-Line 363: EPG is not molecular technique.

Response 23: We have changed to “the small components or a granular level using techniques” on page 13 line 373.

Comments 24-Line 380: Numbers of cited references should be in increasing order. Check this in the whole text.

Response 24: We have checked the cited references in the whole text. However, some cited references should not be in increasing order because they have been cited in the text.

MATERIAL AND METHODS

Comments 25-Lines 401-403: Please, rewrite the sentence. What is “free-choice no-choice conditions”? Maybe you can put “and” or “/” between these two tests. Mention that these tests were performed with insects and that biophysical factors refer to sugarcane traits.

Response 25: Agree. We added “,” in lines 415-417. “When the plants were 12 weeks old, the plants (5–6 leaf stage) were then transported to the laboratory for tested free-choice, no-choice conditions and biophysical factors of sugarcane cultivars”

Comments 26-Line 478: I suppose that average values were compared between (not within) cultivars.

Response 26: Agree. We changed to between the cultivars on page15 line 492

REFERENCES

Comments 27-Line 574: Replace “Matsumuratettixhiroglyphicus” with “Matsumuratettix hiroglyphicus”.

Response 27: Agree. We changed it to “Matsumuratettix hiroglyphicus” in References
